# Evidence of Former Sea Levels from a Passive Seismic Survey at a Sandy Beach; Perranporth, SW England, UK

**Andres Payo** [1,*] **, Gareth O. Jenkins** [1] **, Dave Morgan** [1] **, Nieves G. Valiente** [2] **and Timothy Scott** [3]

1 British Geological Survey, Nicker Hill, Keyworth, Nottingham NG12 5GG, UK; gjenkins@bgs.ac.uk (G.O.J.); djrm@bgs.ac.uk (D.M.)
2 Met Office, Fitz Roy Road, Exeter EX1 3PB, UK; nieves.valiente@metoffice.gov.uk
3 Coastal Processes Research Group, School of Biological and Marine Sciences, University of Plymouth, Plymouth PL4 8AA, UK; timothy.scott@plymouth.ac.uk
* Correspondence: agarcia@bgs.ac.uk; Tel.: +44-0115-936-3103

**Abstract:** Since the end of the last glaciation, the United Kingdom's land surface has been altered by isostatic rebound, rising in the north and sinking in the south. Numerous studies have been published documenting the impact of isostatic rebound on relative sea levels. However, due to the difficulties in acquiring evidence to prove former sea levels, locally, these data can be sparse or absent. In this work, we explored the suitability of the passive seismic survey (PSS) method to estimate the contemporaneous beach thickness in coastal environments where there is a high impedance contrast between the beach deposits and the underlying wave-cut platform. We conducted a three-day survey at Perran Beach, Cornwall, collected 149 measurements using PSS, and interpreted the observations supported by auxiliary topographical, geological, and independent geophysical observation in the study area. The study site is a contemporaneous beach mostly composed of sand underlain by a wave-cut platform composed of igneous and sedimentary rock, therefore high impedance contrast with the sandy beach is anticipated. The elevation of the bedrock relative to the topographical elevation suggests that the bedrock elevation is $-15 \text{ m} \pm 5 \text{ m}$ below the present day mean sea level, which is coherent with the observation of relative sea level rise along the region of the south-west. The present study contributes to our current limited understanding of land and sea level movements by providing further subsurface information to the coastal geological archive of south-west England, a region currently in need of more data to reconstruct land- and sea-level movements.

**Keywords:** geology; isostatic rebound; sea level rise; beach thickness

## 1. Introduction

As the last glaciers retreated from the UK landmass at the end of the last glaciation (c. 9000 years BP), the lithosphere has responded to the removal of glacial loading through a process known as isostatic rebound. In the UK, this movement is most pronounced in SW England. Evidence for this rebound can be found in the form of geomorphological features such as raised beaches, cliff notches, fossil wave-cut platforms, and peat deposits [1]. One of the most established methods for reconstructing past sea levels involves the identification of sea-level index points (SLIPs) [2]. This method utilises the dating of peats that were formed under a tidal influence. However, due to the nature of the deposits, consolidation and/or compaction is an issue when using SLIPs. This has resulted in confusing results that suggest SW England is not subsiding as quickly as previously thought [3]. The geomorphology of the coastline in SW England is predominantly composed of cliffs, beaches, and pocket bays with very limited low-lying areas conducive to peat deposition. As a result, the number of SLIPs in the region is low. There are only 18 scattered SLIPs along the channel coasts of Devon and Cornwall. Information on the indicative meaning of these points is commonly lacking in detail and many data points cannot be accurately related to a former sea level. A more ubiquitous geomorphological feature that can be utilised to help reconstruct past

sea levels is a wave-cut platform. These provide physical geomorphological evidence of an erosion surface formed under past sea levels.

Wave-cut platforms (also known as abrasion platforms) are gently sloping consolidated ledges that extend from the high-tide level at the base of a cliff to below the low-tide level (Figure 1). They are formed as a result of wave abrasion. A platform is broadened as waves erode a notch at the base of the sea cliff, which causes overhanging rock to fall. Over time, under stable or slowly rising sea levels, landward erosion of the cliff results in the formation of a submarine ledge known as a wave-cut platform. Wave-cut platforms are often covered by a beach, which is a layer of loose material (mostly sand and gravel) that is more readily transportable by wind and nearshore waves and currents than the consolidated wave-cut platform material. The challenge is how to measure the elevation of the wave-cut platform surface when it is covered by the beach.

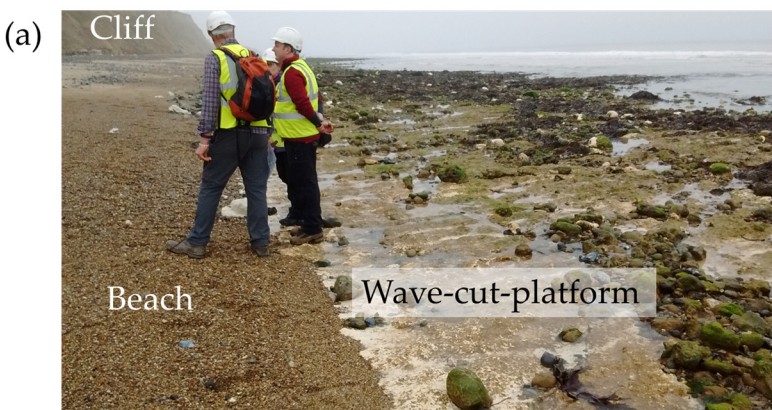

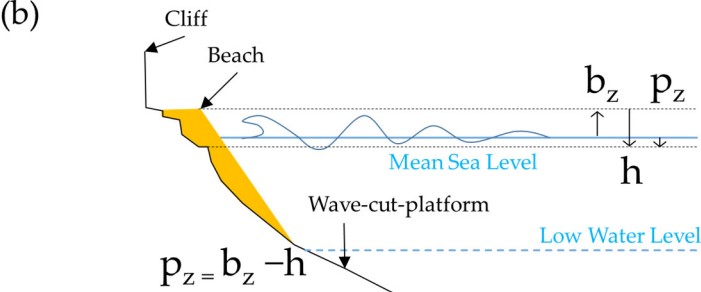

$p_z$ = elevation of wave-cut-platform
$h$ = beach thickness (from PSS data)
$b_z$ = elevation of beach top surface (from topographical survey)

**Figure 1.** Wave-cut platforms are often covered by beach deposits and its position can be obtained from beach thickness and beach elevation: (**a**) photo taken near Trimingham, East England showing a cliff, beach, and wave-cut platform (source BGS); (**b**) schematic showing how wave-cut platform elevation can be obtained from beach thickness and beach elevation observation.

When a beach is present, measuring the location of the wave-cut platform is equivalent to measuring the beach thickness; to convert from beach thickness to a wave-cut platform surface vertical location, we only need an additional observation of beach top surface vertical position (Figure 1). Despite the importance of knowing the beach thickness to assess the future coastal change, there are only a few methods reported in the literature. For example, ref. [4] used combined geophysical and geotechnical methods to characterise beach thickness at Easington coast in East England, where the cliffs and sub-beach platform within the study area consist of tills and the composition of the beach is intimately linked with the available sources of sediment, and as such, is characteristic of a typical mixed-sediment, medium- to high-energy beach. They concluded that this beach environment

is highly variable and cannot be simply modelled as a low-conductivity, low-strength, low-stiffness layer overlying a stiffer, higher-conductivity bedrock with higher strength. Ref. [5] explored the use of the PSS method at Happisburgh, East England, which has a similar beach composition to the beach in Easington, and found that the impedance contrast between the wave-cut platform and beach deposits was not high enough to produce good beach thickness observation, but it was good enough to detect the deeper chalk bedrock. The suitability of PSS to estimate beach thickness on other geological coastal environments is largely unexplored.

The aim of this work was to explore the suitability of the PSS method to estimate the contemporaneous beach thickness on coastal environments where there is a high impedance contrast between the beach deposits and the wave-cut platform. The study site was Perran Beach, Perranporth. It consists of a contemporaneous beach composed mostly of sand underlain by a wave-cut platform composed of igneous and sedimentary rock, therefore high impedance contrast is anticipated. By imaging the wave-cut platform beneath the beach at Perranporth, we can provide additional supplementary supporting evidence to reconstruct former sea levels. We conducted a three-day survey, collected ca. 150 measurements using PSS, and interpreted the observations supported by auxiliary topographical, geological, and independent geophysical observations in the study area.

This paper begins with a detailed description of the geology and geomorphology of the study site, the fundamentals of the PSS and the surveyed locations, the topographical survey conducted to characterise the emerged beach and dune topography and the multichannel seismograph survey method used to independently characterise the subsurface. In the Results section, we describe the results of all the PSS measurements and the multichannel seismograph survey obtained. In the Discussion section, we discuss the advantages and limitations of the PSS method to provide additional supporting evidence of former local sea levels on beaches with high impedance contrast between the beach material and the wave-cut platform in areas where, to date, this has been difficult to acquire.

## 2. Materials and Methods

### 2.1. Study Site

Perranporth Beach is located on the north coast of Cornwall, SW England (Figure 2a). The beach is approximately 3.5 km long and faces WNW into the Celtic Sea. Due to its exposed location, the wave fetch at Perranporth is significant, reaching several thousand kilometres across the Atlantic Ocean. The beach is enclosed by the headlands of Ligger Point (North) and Cligga Head (South) with a 1 km section of cliffs located toward the southern end of the bay (Figure 2b). The beach at Perranporth is composed of beach and tidal flat deposits (undifferentiated). Typically, the upper beach consists of well-sorted medium sand and the lower beach of very well-sorted medium sand [6]. An extensive area of blown sand overlies the bedrock landward of the beach (Penhale, Gear, and Reen Sands). The beach is backed by an extensive dune system both in the north and south that is divided by a small headland (Cotty's Point). The beach and sand dunes are underlain by the Trendrean Mudstone Formation (Devonian). It is composed of dark grey, locally black, mudstone with some laminae of siltstone and fine-grained, dark grey sandstone [7]. Thin beds of light grey siltstone and beige fine sandstone are also locally present. The formation is exposed in the headland to the north of the bay, Ligger Point, and the cliffs backing the southern half of the bay (Cetty's Point). The headland (Droskyn Point and Cligga Head) to the south of the bay is formed of the Grampound Formation (Devonian). This consists of laminae and thin beds of siltstone and sandstone [7].

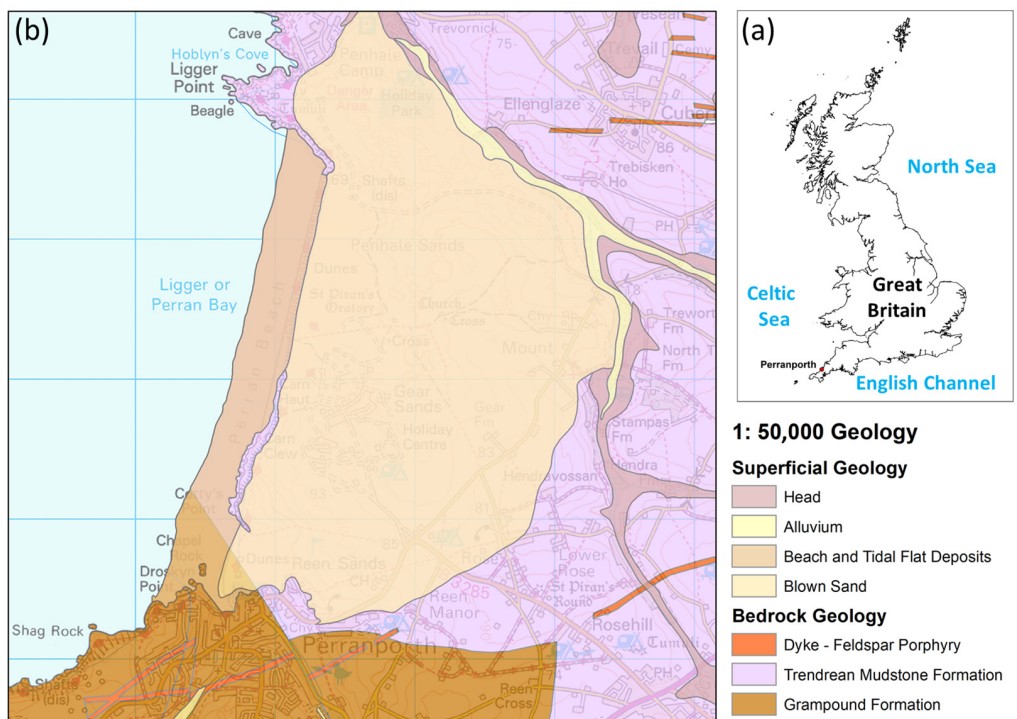

**Figure 2.** Study site location: (**a**) Great Britain High Water Mark; (**b**) Perranporth Beach showing the superficial and bedrock geology of the study area. Map shown in metric scale using British National Grid (blue grid). For reference, the latitude and longitude of Ligger Point is 50°22′48.0″ N, 5°09′21.7″ W.

### 2.2. Field Observations

2.2.1. Passive Seismic Survey: Horizontal to Vertical Spectral Ratio Method

We used the PSS method to estimate the beach thickness and to build confidence in the subsurface lithological model. PSS measures background seismic noise (both natural <1 Hz and man-made >1 Hz) to estimate the thickness of the different lithologies through different time domains and spectral techniques. Seismic tremor, commonly called seismic 'noise', exists everywhere on the Earth's surface. It mainly consists of surface waves, which are the elastic waves produced by the constructive interference of P and S waves in the layers near the Earth's surface. Seismic noise is mostly produced by wind and sea waves. Industries and vehicle traffic also locally generate tremor, although essentially at high frequencies (>1 Hz), which are readily attenuated. PSS consists of a series of single-station point recordings, generally arranged into linear transects. These can be of any length and, where organized into an appropriate grid pattern, can be used to generate 3D surfaces of target horizons. Best results are achieved where independent depth control such as borehole information is available to calibrate the results. For this study, we used the Tromino ENGY-3G (www.moho.world/tromino), a small (10 × 14 × 8 cm), portable (~1 kg), broadband, three-component seismometer and the proprietary software Grilla (v7.0) that implements the Horizontal-to-Vertical Spectral Ratio (H/V) method [8,9]. The reason behind using the spectral noise ratio is that seismic noise varies largely in amplitude as a function of the noise "strength", but the spectral ratio remains essentially unaffected and is tied to local subsoil structure [10]. The Grilla software also provides routines for quality control of the H/V analyses following the European SESAME project directives [11].

Seismic ground noise acts as an excitation function for the specific resonances of the different lithologies in the subsoil. For example, if the subsoil has resonant frequencies of 0.8 and 20 Hz, the background seismic noise will excite these frequencies, making them visible when applying the H/V technique on the recordings and these resonant frequencies can be used as a proxy for cover thickness. In a simple double layer stratigraphy consisting

of sedimentary cover and bedrock, there is a simple equation [12] relating the resonance fundamental frequency $f_0$ to the thickness $h$ of the layer and $Vs$ (the shear wave velocity in the same layer):

$$f_0 = Vs/(4h), \tag{1}$$

where the value of $Vs$ varied for different materials with typical values of 100–180 m/s for clay, 180–250 m/s for sand, and 250–500 m/s for gravel. In case of several peaks on the H/V curve, the peak with the lowest frequency was the fundamental mode ($f_0$, generally the bedrock-cover limit), and other peaks (i.e., $f_1, f_2, \ldots, f_n$) corresponded to other geological limits that also caused seismic motion amplification. We acknowledge here that it is also possible to estimate $h$ from the empirical regression model between $f_0$ and $h$ (e.g., [13]) but for this study area, the lack of $h$ values obtained from alternative non-seismic survey methods prevented us from using this empirical approach.

For the stations located on the beach at the study site, we would expect to see the fundamental peak corresponding with the interfaces between the contemporaneous beach deposits and the bedrock ($f_0$). There are no boreholes located in or close to the study area to provide information about depth to bedrock. Table 1 shows different values for $f_0$ for different beach thickness and $Vs$ values. The Tromino was set up to measure background noise for a duration of 8 min at 128 Hz sampling frequency per station of point of interest (POI). According to the Nyquist frequency, the highest frequency that can be recovered from a digitised signal is always lower than half the sampling frequency [14]. Hence, when sampling at 128 Hz, one can resolve signals at frequencies at most 64 Hz high. For a sand beach deposit (max $Vs$ ~250 m/s) with a thickness O(10m), we will expect the $f_0$ peak to be around 6.25–12.5 Hz, which is well within the maximum observable frequency when sampling at 128 Hz. In practice, spectral estimates are statistical in nature and to have stable results, the observation time should be long enough to comprise at least 10 repetitions of the longest period of interest. For a beach thickness of 10 m, the longest period ($T_0$) (i.e., lower frequencies derived from Table 1) corresponded with $f_0$ ~2.5 Hz ($T_0 \cong 1/2.5 = 0.4$ s), which means that a time series of 4 s in length (i.e., $10 \times 0.4$ s) will be long enough to capture the lowest frequency peaks. Because we are going to extract information from seismic noise, we expect fluctuations with time, which can be appropriately controlled by sampling a number N of 4 s windows sufficient to compute an average that is statistically significant. Common practice shows this number N to be 30–50, which means that in the above example, a total recording time of $50 \times 4$ s = 200 s = 3.33 min. A total length of 3.3 min could have been enough, but in anticipation that a significant amount of the time series might be contaminated by anthropogenic noise (i.e., beach goers), we used a duration of 8 min.

**Table 1.** Estimated fundamental frequencies using Equation (1) for different beach thickness and $Vs$ values.

| Thickness (m) | Clay [1] | Sand [2] | Gravel [3] |
|---|---|---|---|
| 1 m | 25–45 Hz | 45–62.5 Hz | 62.5–125 Hz |
| 10 m | 2.5–4.5 Hz | 4.5–6.25 Hz | 6.25–12.5 Hz |

[1] $Vs$ = 100–180 m/s; [2] $Vs$ = 180–250 m/s; [3] $Vs$ = 250–500 m/s; from "The Short TROMINO how to ENG.pdf" v1.1 by www.moho.world/tromino.

A three-day field survey was conducted starting on 21 August 2017. A total of 149 measurements were measured along ten lines perpendicular to the shoreline (Figure 3). On each line, the sampled locations were equally spaced (every 30 m) from the cliff toe to the seaward limit of the dry beach at the time of the survey. The survey was conducted by a three-person team. The points of the stations were located using a hand-held Garmin GPS (GPSMAP 64s) on which the planned locations were pre-loaded as way-points. During the survey, one of the team members was in charge of navigating to the way-points and marking with pegs the approximate locations. The other two team members, each equipped with a Tromino station, were in charge of recording with the Tromino. The Tromino unit was

coupled to the ground at each point using three 6 cm long spikes (~4 cm penetration length). The coupling was achieved by alternately pressing on the lowermost corners of the box and on the middle of the top edge to set the Tromino in a horizontal position, using the spirit level on the Tromino to ensure the equipment was level. Caution was taken to avoid creating any non-coherent noise while recording.

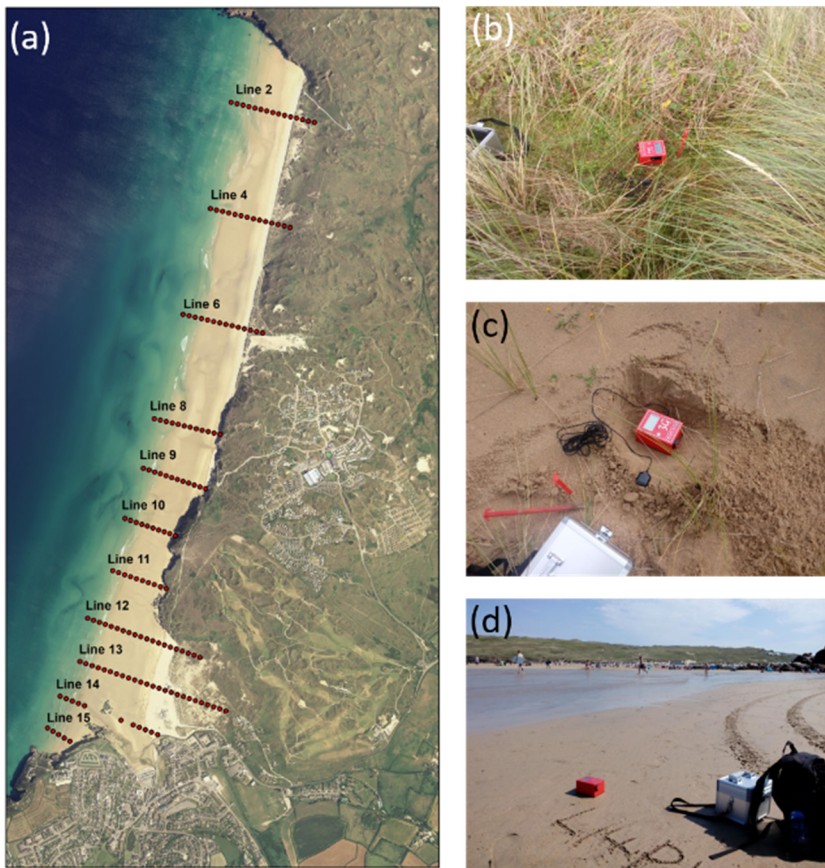

**Figure 3.** (**a**) Location of the 149 stations sampled using the passive seismic survey method at Perran Beach where each station was identified by the line (L) and point (P) number (e.g., L12P1); (**b**,**c**) physical setup at stations located at the vegetated and non-vegetated dune; (**d**) at the intertidal beach.

2.2.2. Estimation of Surface Wave Velocity Using a Multichannel Seismograph Survey Method

The propagation velocity (called phase velocity) of surface waves is frequency (or wavelength) dependent (this property is called dispersion). The dispersiveness of soils is determined mainly by the vertical variation in $Vs$. By recording fundamental-mode Rayleigh waves propagating horizontally and directly from the seismic source to the receiver, the dispersive properties directly beneath the source and geophone spread can be measured and usually represented by a curve (called the dispersion curve) depicting the variation of phase velocities with frequency. This curve is then used to estimate the vertical variation of $Vs$ (called 1-D $Vs$ profile) through a process called inversion. Although generally thought to be a relatively easy seismic method to use, several complications may interfere with the effectiveness of the surface-wave method, especially if improper acquisition or processing (or both) techniques are used [15].

We used the multichannel analysis of surface waves (MASW) method [16], which utilises pattern-recognition techniques made possible by the multichannel recording and processing approaches [17]. MASW employs multiple receivers (geophones) equally spaced along a linear survey line with seismic waves generated by an impulsive source (e.g., a

person jumping) and propagated along the receiver line where they are recorded synchronously. This approach allows for recognition of the various propagation characteristics of the seismic wave field. Multiple receivers are equally spaced along a linear survey line (Figure 4a). Considering the frequency-with-depth dependency of surface waves and the response characteristics of geophones, low-frequency (4.5 Hz) geophones are normally used as receivers and a heavy impact seismic source such as an adult person jumping can produce a broadband, relatively low-frequency signal. Distance between source and the nearest receiver-station (called source offset) was chosen to minimise near-field effects caused by excessive stress–strain relationships from the impact source [18]. This source offset is usually chosen to be about the same as the maximum depth of investigation. However, situations do exist where source-to-receiver offsets less than the depth of interest are appropriate [19]. Receiver spacing was chosen to avoid any possible spatial aliasing of the shortest wavelength recorded and to maximise the effectiveness of dispersion analysis [18]. Total length of receiver spread determines the farthest offset and receiver spacing and needs to be short enough that strong body and higher-mode surface waves, usually dominant at far offsets, do not interfere with fundamental-mode dispersion curve analysis. Specific source-receiver (SR) configurations are defined by source offset, receiver spacing, and receiver spread. A measurement is a multichannel record (called a shot gather) resulting from one or more seismic impact(s) recorded at a single point by a fixed receiver spread. The vibration history related to the source-generated seismic wave field was measured by each geophone within the spread and is represented by a time series referred to as a trace. From each measurement comes a single 1-D *Vs* profile of the earth materials directly beneath the spread.

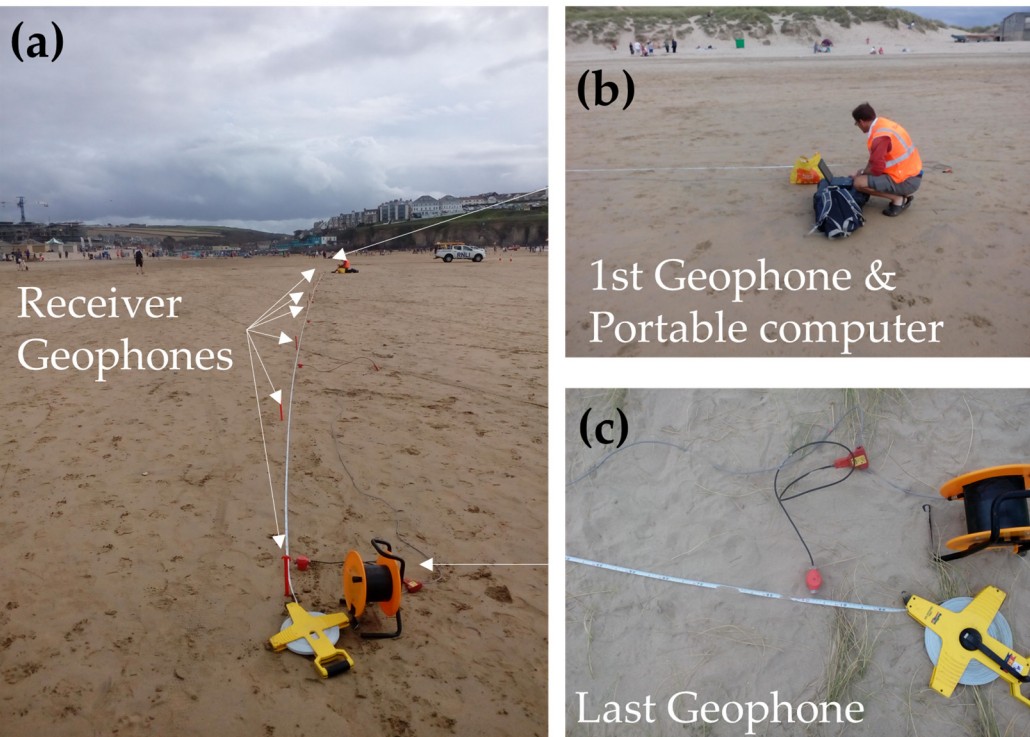

**Figure 4.** Photos (source BGS) illustrating a typical MASW field configuration showing the chain of receiver geophones (**a**) and details of first (**b**) and last geophone (**c**).

We used a chain of nine vertical geophones (SoilSpy Rosina) with a receiver spacing of 5 metres, resulting in a total length of 40 metres and the surface waves phase velocity spectra module in Grilla software (v7.0) for processing. We conducted two MASW surveys on 8 September 2017, first on the beach (on 23 August 2017 starting at 17:12) and then on the dune (on 23 August 2017 starting 17:36), as shown in Figure 5. The beach survey was

conducted on the intertidal and the sand was still wet from the last high tide of the day. The dune transect was conducted along a path free of vegetation, as shown in Figure 5. The single station measurements points 6, 7, and 8 on Line 12 and points 13, 14, and 15 on Line 13 were the closest to the beach transects and about 160 m away from the transects' centre location.

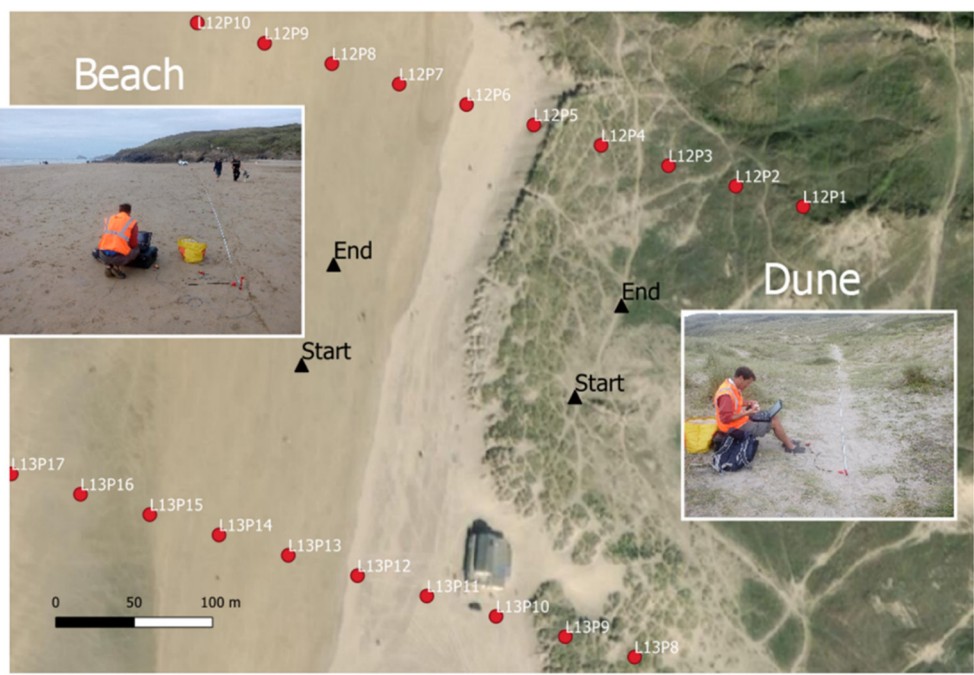

**Figure 5.** Location of the MAWS set up on the beach and dune. Pictures labelled as beach and dune show the field survey setup at Perran Beach on 23 August 2017 (source BGS). The location of the first and ninth geophones are indicated by the black triangles and labelled as start and end, respectively. Source of aerial imagery is the ArcGIS World Imagery accessed on 13 August 2020: Esri, Maxar, GeoEye, Earthstar Geographics, CNES/Airbus DS, USDA, USGS, AeroGRID, IGN, and the GIS User Community.

2.2.3. Estimation of Elevation of Bedrock Relative to Present Mean Sea Level

As illustrated in Figure 1, to obtain the elevation of the wave-cut platform, $p_z$, relative to present mean sea level (MSL), we need both the beach thickness, $h$, and the beach elevation, $b_z$, relative to MSL.

$$p_z = b_z - h \tag{2}$$

The beach thickness, $h$, was obtained directly from the PSS observation and Equation (1). For this study, we used the Ordnance Datum Newlyn (ODN) as proxy for present day MSL. The beach elevation, $b_z$, relative to ODN was extracted at each sampled location from the digital elevation model (DEM) created from topographic surveys conducted over the summer of 2017. The complete DEM covered the inter- and supra-tidal beach as well as the dunes.

Two sets of topographic data were used: (1) 3D Real Time Kinematic (RTK) GPS/GNSS mounted on an All-Terrain Vehicle (ATV) survey; and (2) photogrammetric data collected from an unmanned aerial vehicle (UAV). The photogrammetric data covered the south and north dunes. This was collected using a DJI Phantom 4 quadcopter, covering the supratidal zone up to an elevation of 30 m ODN. Ground control points (GCPs) were vertically and horizontally distributed throughout the survey region and surveyed by RTK-GPS for constraining bundle adjustment during the post-processing workflow using the commercial software Agisoft. The inter-tidal and supratidal zone was surveyed using an ATV-based Trimble 5800 RTK-GPS with line spacing of 20–25 m. Topographic surveys were carried out using 1 Hz continuous measurements. A digital DEM for the inter-

and supratidal was constructed using Loess interpolation function [20], with a maximum permissible interpolation error level of 0.15 m [21,22]. This dataset was then combined with the photogrammetric data for the final DEM construction using natural neighbour interpolation function [23].

## 3. Results

### 3.1. Weather and Environmental Conditions during the Survey

On the three days of the survey, the weather conditions were mild, with average temperatures ranging between 17.5 and 19.1 °C (Newquay Weather Station). Average wind speeds varied between 4.1 and 7.8 mph, blowing from a westerly direction on 21 and 23 August and from the east on the 22 August. Average gust speeds ranged between 6.0 and 10.5 mph, with a maximum gust speed of 24.2 mph measured at Porth in Newquay on 22 August. Significant wave heights during the survey period (taken from the Perranporth Waverider buoy) ranged between 0.7 and 1.2 m with a dominant wave period between 7 s and 13 s. As the data were collected during the summer holiday season, the beach was busy with frequent footfall from passing beach goers (Figure 6a). Dogs were often attracted to the observation location while sampling and some non-coherent noise is likely to be observed on the records. In Perranporth, beach users deployed wind breakers using a hammer and pegs to fix it in the sand (Figure 5b). Field surveyors were wearing a high visibility jacket and instructed beach goers not to walk too close to the unit while sampling to minimise the non-coherent noise signal.

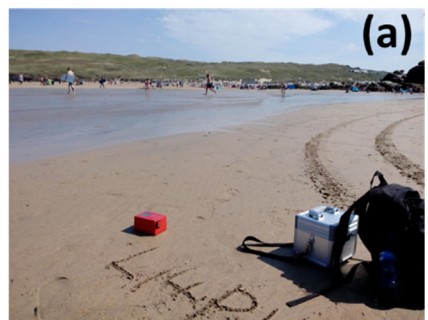
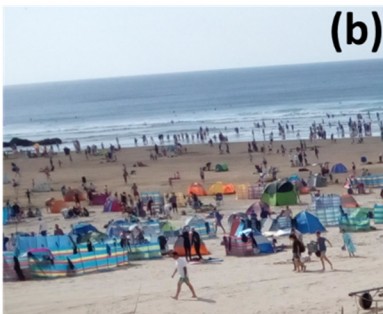
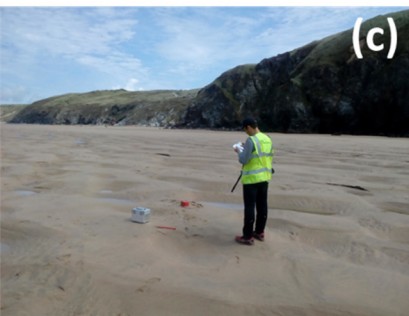

**Figure 6.** Field survey was conducted in the summer season: (**a**) shows typical distance between beach users and Tromino unit, (**b**) portable wind breakers deployed by beach users during the survey, and (**c**) member of staff using a high visibility jacket to indicate location and indicate beach users not to get too close to the unit while sampling (Source BGS).

### 3.2. MASW 1-D Vs Profiles

Figure 7 shows the observed and modelled surface wave velocity dispersion curves at the beach and in the dunes. The maximum depth of investigation can be estimated from the recorded phase velocity at which the minimum frequency of the fundamental mode on the dispersion curve is clearly defined. The fundamental frequency mode was identified as the curve with the lowest velocity values for each frequency. For the beach record, the minimum frequency was approximately 8 Hz and the corresponding velocity was around 400 m/s. By dividing 400 m/s by 8 Hz, we obtain 400 m/s/8 Hz = 50 m. This is the maximum Rayleigh wavelength ($\lambda_{max}$) that can be recognised from this survey. The maximum wavelength of investigation was not the maximum depth of investigation. Rayleigh waves in fact induce the major part of the ground displacement at depths between 1/3 and 1/2 of their wavelength. It is therefore generally assumed that the velocity at which a Rayleigh wave is moving refers to a depth between $\lambda_{max}/3$ and $\lambda_{max}/2$. In this case, by dividing the maximum wavelength by 2, we had an estimation (by excess) of the maximum depth of investigation achieved by the survey as $\lambda_{max}/2 = 50/2 = 25$ m depth. The modelled Rayleigh wave velocity dispersion curve (Figure 7a) was obtained assuming the 1D *Vs* profile shown in Figure 7a, suggesting that the *Vs* was 220 m/s for the first 10 m,

then $Vs$ = 500 m/s until 25 m depth (i.e., first layer was 10 m thick and second layer was 15 m thick). The average $Vs$ could be obtained as (10/25) × 220 + (15/25) × 500 = 388 m/s for the beach top layer. For the dune survey, the minimum frequency was approximately 10 Hz and the corresponding $Vs$ was around 400 m/s. By dividing 400 m/s by 10 Hz, we obtained $\lambda_{max}$ = 400 m/s/10 Hz = 40 m and the maximum depth of investigation achieved by the survey was $\lambda_{max}$/2 = 40/2 = 20 m. The modelled Rayleigh wave velocity dispersion curve (Figure 7b) was obtained assuming the 1D $Vs$ profile shown in Figure 7b, from which it can be inferred that the bedrock interface was around 23 m depth and close to the depth of investigation. The average $Vs$ could be then obtained as (4/23) × 230 + (9/23) × 300 + (10/23) × 500 = 374 m/s for the dune top layer. It was observed that the differences between the averaged $Vs$ obtained for the beach and dune (374 m/s + 388 m/s)/2 = 381 m/s was less than ± 2%.

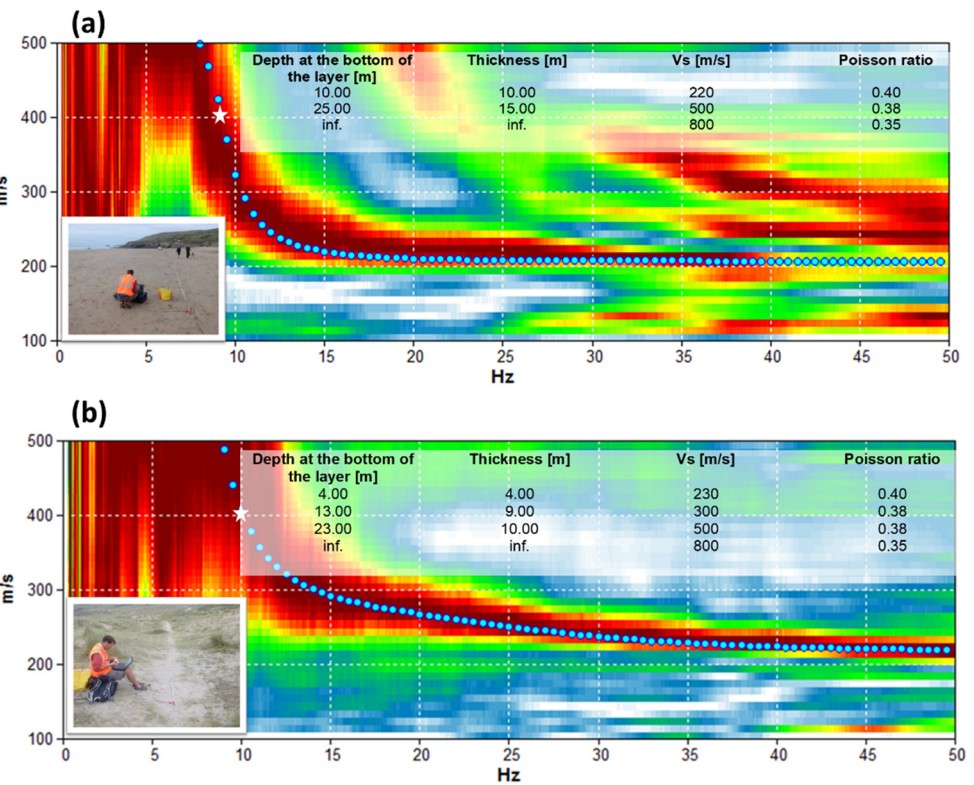

**Figure 7.** Phase velocity dispersion observed in the beach (**a**) and dunes (**b**) at Perran Beach on 23 August 2017. For a specified frequency, high spectral content areas are shown in red colour shades, with low spectral content shown in blue. Blue circles represent the modelled Rayleigh wave phase velocity dispersion curve. The main model parameters are indicated as a table. Thickness = inf represents the bedrock. The white star indicates the minimum frequency used to estimate the maximum depth of investigation.

### 3.3. Single Station Passive Seismic Survey

The quality and clarity of the recorded passive seismic records varied significantly (Figure 8). We computed the H/V spectral ratio using the entire 8 min record (i.e., not eliminating the non-coherent noise), spectra were calculated at 20 s triangular windows with 5% smoothing and the horizontal component shown is the average of the horizontal east and north components. Through visual inspection, the data from the 149 observation stations could be clustered into four categories based on the coherence of the noise recorded and the clarity of the fundamental frequency on the H/V spectral ratio. The description and percentage of each type can be summarised as:

- **Coherent strong** (32%)—Clear H/V time history signal with low levels of noise interference. Strong horizontal to vertical spectral ratio peak.

- **Coherent weak** (36%)—Slightly noisy H/V time history plot, but still evidence of a clear horizon. Weak horizontal to vertical spectral ratio peak.
- **Non-coherent strong** (11%)—Noisy H/V time history plot with slight evidence of a horizon. Weak to very weak horizontal to vertical spectral ratio peak.
- **Non-coherent weak** (21%)—Very noisy signal with no evidence of a horizon. No horizontal to vertical spectral ratio peak.

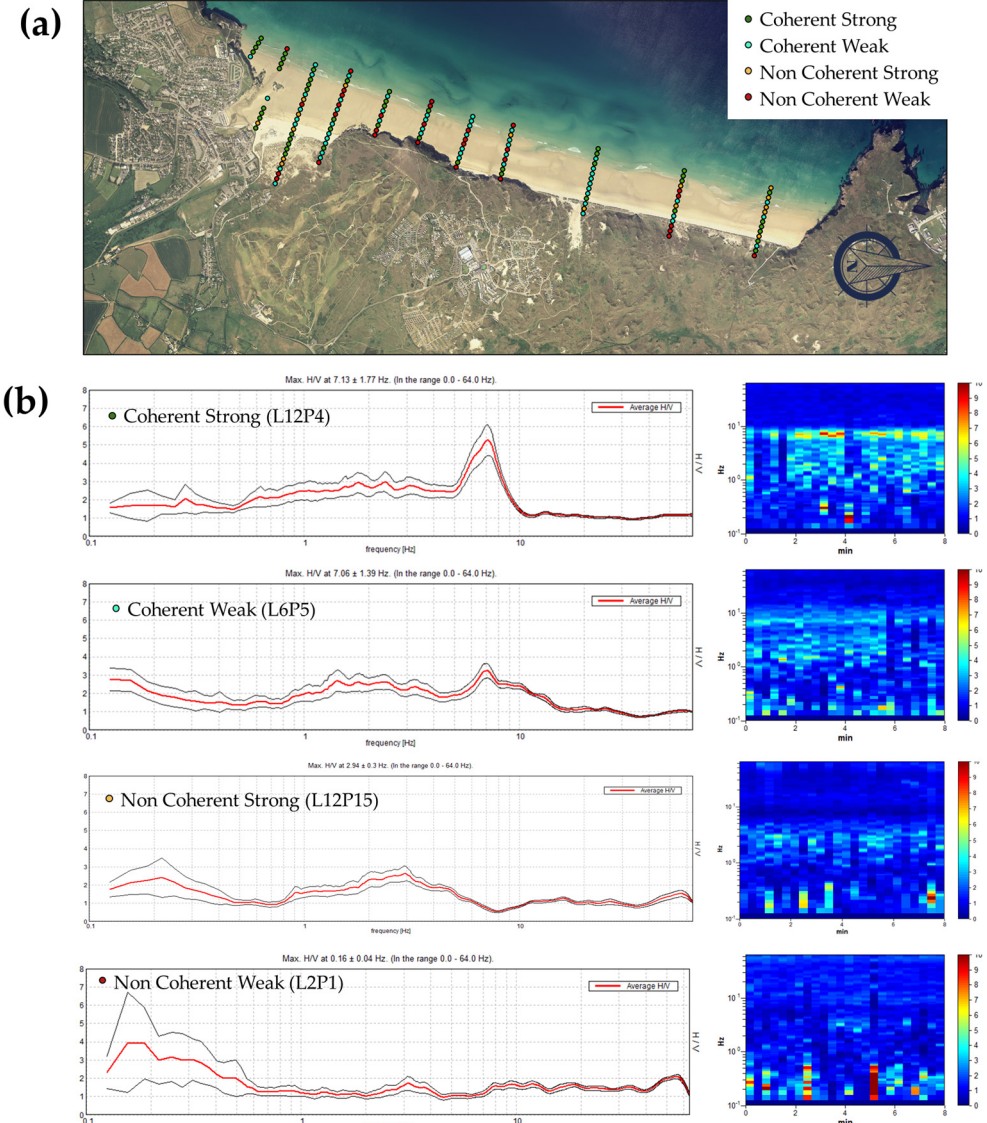

**Figure 8.** Summary of the 149 recorded single station measurements illustrating the (**a**) spatial distribution of the Tromino sample points displaying the four quality categories of the data collected and (**b**) representative H/V time history and horizontal to vertical spatial ratio plots for each of the four categories.

Figure 8a shows the spatial distribution of the four identified categories. The non-coherent weak data appeared to correspond with either the dunes, dry beach, or lower beach. The measurements taken on the dunes need to be interpreted with caution as vegetation was present (e.g., Marram grass), which can cause non-coherent-noise signal, even under low wind conditions, and poor sensor to ground coupling. It was also notable that a number of non-coherent weak records occurred at the end of each transect near the low water mark. This is believed to be mainly due to the saturated nature of the sediment, which made it difficult to securely couple the Tromino instrument in place. Interference

from waves breaking just offshore, or again possibly due to bathers/people who tend to congregate/walk along the edge of the water line, could also have had a negative effect on the quality of the data. Another factor that may cause poor data returns is wind, which was relatively high (gusts up to 24.2 mph) for the duration of the survey.

Figure 9 shows the observed and modelled H/V curves for the three closest stations to the MASW survey conducted at the dunes. These points were points 2, 3, and 4 of line 12, which were about 150 m away from the centre of the array of geophones at the dune. All three single component spectra showed the expected decrease on the vertical component relative to the horizontal components at around 5 to 7 Hz. The H/V spectra for points 3 and 4 showed a clear maximum at 5.91 Hz and 7.22 Hz, and on point 2, a less clear maximum at 5.78 Hz. Assuming a simple two-layer model and using the average $Vs$ = 374 m/s velocity, we estimated that these peaks corresponded with depths of 14 m to 18 m. Topographical elevation at points 2, 3, and 4 were 25 m, 18.6 m, and 13.5 m, respectively. Elevation of the MASW survey on the dunes was similar to the one at point 4. Table 2 shows the estimated depth to bedrock using the elevation at point 4 as a vertical datum, which suggests that the bedrock is not horizontal, but has a positive slope inland as platform elevation increases from −0.5 m to 7 m relative to ODN.

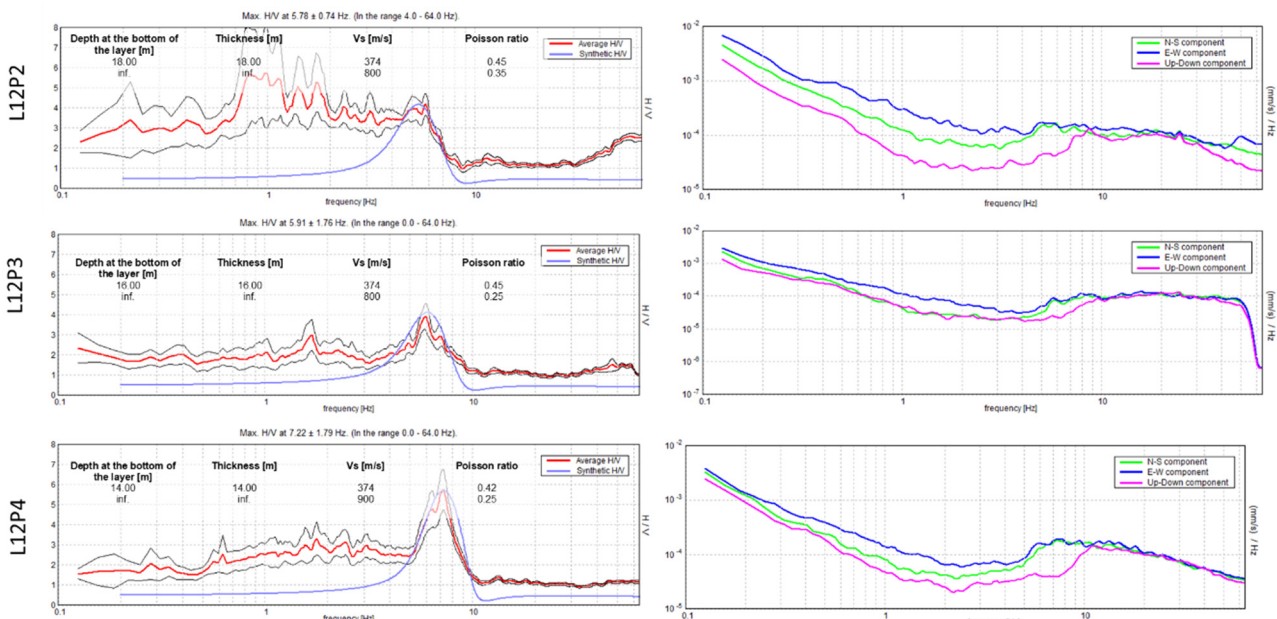

**Figure 9.** Single station H/V observed and modelled at the three nearest stations to the dune MASW location. The labels L12P2-4 indicate the points 2, 3, and 4 on line 12. The right panels show the single component spectra as solid blue, green, and magenta lines. The left panels showed the averaged H/V as a solid red line, with the standard deviation as thinner black lines. The modelled or synthetic spectra are shown as a blue line and the main model parameters are shown as text.

**Table 2.** Estimated elevation of the wave-cut platform calculated from beach elevation minus beach elevation at points 2, 3, and 4 on line 12.

| Point | Modelled Beach Thickness, $h$, (m) | Beach Elevation, $b_z$, (m) | Elevation of Wave-Cut Platform [1], $p_z$, (m) |
|---|---|---|---|
| L12P2 | 18 | 25 | 7 |
| L12P3 | 16 | 18.6 | 2.6 |
| L12P4 | 14 | 13.5 | −0.5 |

[1] Calculated as $p_z = b_z - h$.

Figure 10 shows the observed and modelled H/V curves for the six closest stations to the MASW survey conducted at the beach (three on line 12 and three on line 13).

These points were points 6, 7, and 8 on line 12 and points 13, 14, and 15 on line 13, which were about 150 m away from the centre of the array of geophones at the beach. All six single component spectra showed the expected decrease in the vertical component relative to the horizontal components at frequencies from 4 to 7 Hz. Assuming a simple two-layer model and using the average $Vs$ = 388 m/s velocity, we estimated that the fundamental peaks corresponded with a layer thickness from 23 m to 16 m. Elevation differences between all six points were less than 1.8 m. The estimated depth to bedrock (Table 3) along line 13 varied between −15.6 and −13.9 m ODN while along line 12 varied between −12.1 m and −20.6 m ODN.

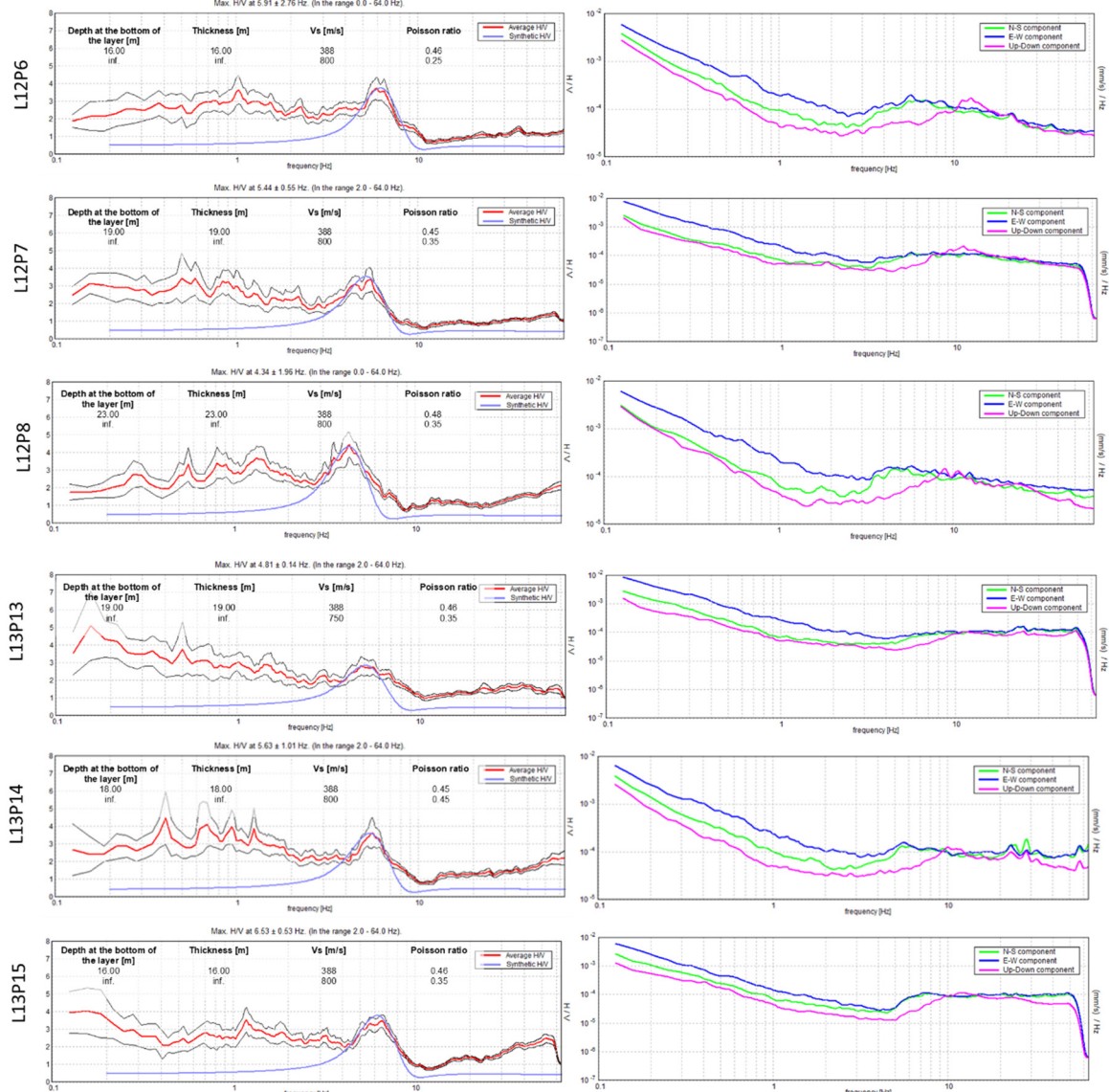

**Figure 10.** Single station H/V observed and modelled at the six nearest stations to the beach MASW location. From top to bottom, the labels LIPJ indicate points 6, 7, and 8 on line 12 and points 13, 14, and 15 on line 13. The right panels show the single component spectra as solid blue, green, and magenta lines. The left panels show the averaged H/V as a solid red line, with the standard deviation as thinner black lines. The modelled or synthetic spectra are shown as a blue line and the main model parameters are shown as text.

**Table 3.** Estimated elevation of wave-cut platform from beach elevation and modelled beach thickness at the six nearest points to the beach MASW survey.

| Point | Modelled Beach Thickness, $h$, (m) | Beach Elevation, $b_z$, (m) | Elevation of Wave-Cut Platform [1], $p_z$, (m) |
|---|---|---|---|
| L12P6 | 16 | 3.9 | −12.1 |
| L12P7 | 19 | 3.1 | −15.9 |
| L12P8 | 23 | 2.4 | −20.6 |
| L13P13 | 19 | 3.4 | −15.6 |
| L13P14 | 18 | 2.7 | −15.3 |
| L13P15 | 16 | 2.1 | −13.9 |

[1] Calculated as $p_z = b_z - h$.

### 3.4. Estimated Beach Thickness and Elevation of Wave-Cut Platform

In Appendix A, we show for each one of the 149 locations sampled with the PSS method, the coordinates (in British National Grid coordinate system or EPSG:27700-OSGB 1936), the peak frequency and standard deviation of the peak frequency, the $Vs$ value that best fits the peak frequency, and the estimated beach thickness resulting from applying Equation (1). The fundamental frequency for all 118 measurements for which it was possible to identify the peak varied from 2.59 Hz to 24.5 Hz with a median value of 5.91 Hz, mean value of 6.25 Hz, and standard deviation of ±2.89 Hz.

Figure 11 shows the depth to bedrock histogram and map for 118 locations from which we have been able to estimate the fundamental frequency on the H/V spectra (i.e., the locations for which the signal was of type NCW were not included). The depth to bedrock varied between 4 m and 37 m, with the median value of 16 m, mean value of 18 m, and standard deviation of ±7 m. Depth to bedrock across most of the transects (lines 4, 6, 8, 9, 10, 11, and 15) was most shallow close to the cliff or dune edge, with depth to bedrock increasing toward the low water mark. Values along lines 12, 13, and 14 showed no clear tendency, with the largest values not being the closest to the lower water mark. Depth to bedrock along line 15 (i.e., the most southern line) showed consistently low values while line 2 (i.e., the most northern line) showed consistently large values.

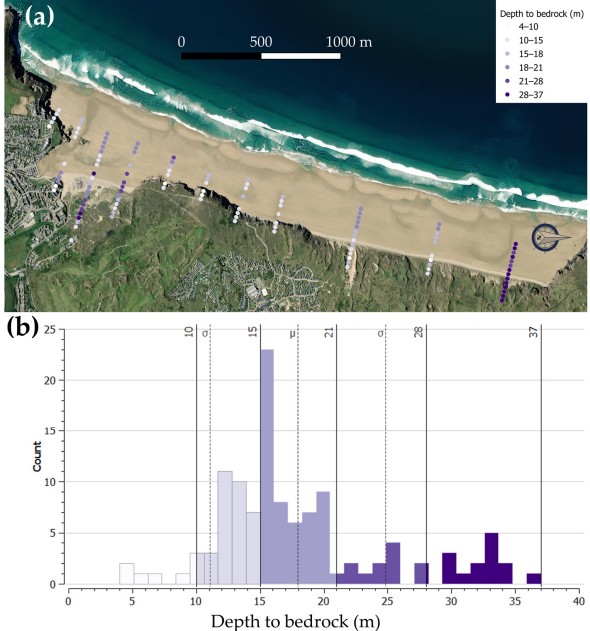

**Figure 11.** Depth to bedrock frequency spatial (**a**) and frequency distribution (**b**) obtained from the PSS sampled at Perranporth Beach. The vertical lines in (**b**) represent the max and min value for each colour group.

Figure 12 shows the elevation of the bedrock histogram and map for 118 locations from which we were able to estimate the fundamental frequency. The bedrock elevation varied between −33 m and 20 m, with the median and mean value of −15 m and standard deviation of ±10 m. Elevation of bedrock across most of the transects (lines 4, 6, 8, 9, 10, 11, and 15) showed a tendency to decrease with the distance to the cliff or dune edge with the lower bedrock elevation values closer to the low water mark. Elevation of bedrock values along lines 12, 13, and 14 showed no clear correlation with lower elevation values being closer to the lower water mark. Bedrock elevation line 15 (i.e., the most southern line) showed the majority of values within one standard deviation above the mean value while line 2 (i.e., the most northern line) showed the majority of values lower than the mean value minus one standard deviation.

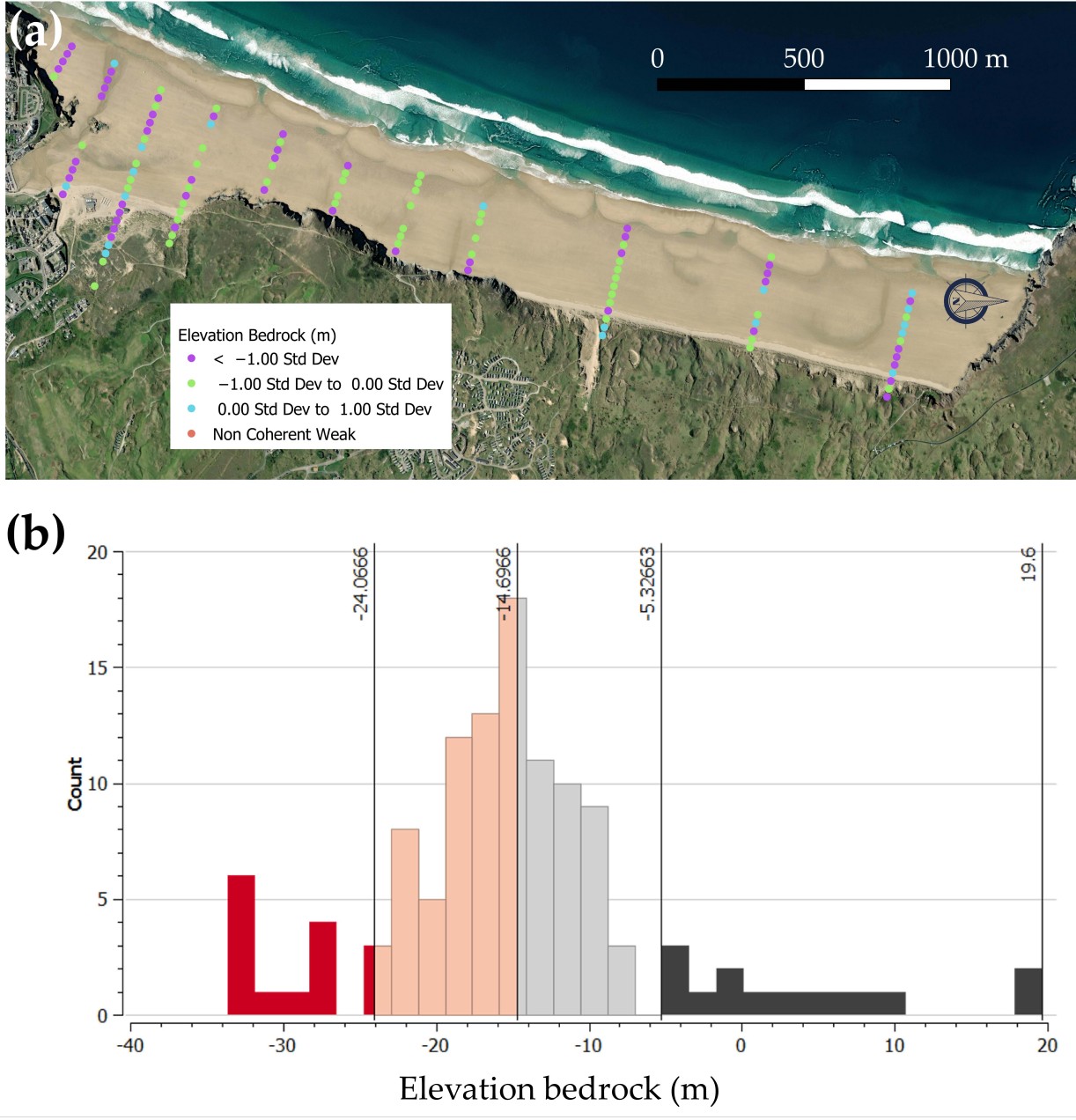

**Figure 12.** Elevation of the bedrock spatial (**a**) and frequency distribution (**b**) obtained from the PSS sampled at Perranporth Beach. The vertical lines in (**b**) represents the max and min value for each colour group.

## 4. Discussion

In this work, we explored the suitability of the PSS method to estimate the beach thickness on coastal environments where there was a high impedance contrast between the beach deposits and the wave-cut platform. The selected study site for this investigation was the 3.5 km long Perran Beach, where contemporaneous beach deposits were composed mostly of sand and the bedrock consisted of igneous and sedimentary rock, and therefore high impedance contrast was anticipated.

We have shown how due to the small size of the PSS unit used (less than 1 kg of weight), a team of three people using two PSS units were able to collect 149 measurements over a three-day survey. We found a strong H/V signal on 68% of the measurements, a weak H/V signal for 11%, and non-identifiable H/V signal for 21% of the measurements. The lack of a clearly identifiable H/V peak for 32% of the measurements is believed to be mainly due to the saturated nature of the sediment, which made it difficult to securely couple the Tromino instrument in place. Interference from waves breaking just offshore, or again possibly due to bathers/people who tend to congregate/walk along the edge of the water line, could also have had a negative effect on the quality of the data.

From the MASW survey conducted at the beach and assuming a simple two-layer model, we estimated that $Vs = 388$ m/s $\pm 2\%$ uncertainty, which was well within the expected range of 250–500 m/s for gravel-dominated deposits. The fundamental frequency for the 78% of the measurements for which it was possible to identify the peak, varied from 2.59 Hz to 24.5 Hz. The elevation of the bedrock relative to the topographical elevation suggests that the bedrock elevation is $-15$ m $\pm$ 5 m below the present day mean sea level, which is coherent with the observation of relative sea level rise along the south-west region, risen by $21 \pm 4$ m during the past 9000 years [3].

Imaging of the wave-cut platform provides physical geomorphological evidence which compliments and allows for correlation with existing SLIPs in the region. The data acquired by the Tromino are intended as supplementary to other sea level reconstruction data and should be used in combination with these to ensure a high level of precision to the calculations. Further work could include dating of the wave-cut platform using cosmogenic isotopes, which would reveal when the platform was last exposed subaerially, using techniques outlined by [19], and make calculations more robust.

The present study contributes to our current limited understanding of land and sea level movements by providing further subsurface information to the coastal geological archive of south-west England, a region currently in need of more data to reconstruct land- and sea-level movements.

**Author Contributions:** Conceptualization, A.P.; Methodology, A.P. and D.M.; Software, A.P. and D.M.; Validation, A.P., D.M. and G.O.J.; Formal analysis, A.P. and G.O.J.; Investigation, A.P. and G.O.J.; Resources, A.P. and T.S.; Data curation, A.P. and N.G.V.; Writing—original draft preparation, A.P. and G.O.J.; Writing—review and editing, All; Visualization, A.P., G.O.J. and D.M.; Supervision, A.P.; Project administration, A.P.; Funding acquisition, A.P. and T.S. All authors have read and agreed to the published version of the manuscript.

**Funding:** This work was funded by NERC (http://www.nerc.ac.uk) as part of the BLUEcoast project (https://projects.noc.ac.uk/bluecoast/) (NE/N015649/1).

**Institutional Review Board Statement:** Not applicable.

**Informed Consent Statement:** Not applicable.

**Data Availability Statement:** The data that support the findings of this study are available on request from the corresponding author, AP.

**Acknowledgments:** Andres Payo, Gareth Jenkins, and Dave Morgan publish with the permission of the Executive Director, British Geological Survey (UKRI).

**Conflicts of Interest:** The authors declare no conflict of interest. The funders had no role in the design of the study; in the collection, analyses, or interpretation of data; in the writing of the manuscript, or in the decision to publish the results.

## Appendix A

**Table A1.** For each one of the 149 locations sampled with the PSS method, the coordinates (in British National Grid coordinate system or EPSG:27700-OSGB 1936), the peak frequency and standard deviation of the peak frequency, the *Vs* value that best fits the peak frequency, and the estimated beach thickness resulting from applying Equation (1). Background colour alternates between grey and white to visually separate the different transects.

| Code | Easting (OSGB_36) | Northing (OSGB_36) | Elev. (m) | $f_0$ (Hz) | std (Hz) | Vs (m/s) | Beach Depth (m) | Type |
|------|------|------|------|------|------|------|------|------|
| L2P2 | 176,412.407 | 57,317.144 | 28.59 | 3.13 | 1.68 | 374 | 30 | CS |
| L2P3 | 176,385.159 | 57,323.772 | 18.32 | 3.03 | 0.07 | 374 | 31 | CW |
| L2P4 | 176,357.912 | 57,330.401 | 9.83 | 2.81 | 0.12 | 374 | 33 | CS |
| L2P5 | 176,330.665 | 57,337.029 | 4.19 | 2.59 | 0.3 | 388 | 37 | NCS |
| L2P6 | 176,303.417 | 57,343.657 | 2.71 | 2.97 | 0.09 | 388 | 33 | CS |
| L2P7 | 176,276.17 | 57,350.286 | 1.81 | 2.91 | 1.08 | 388 | 33 | CS |
| L2P8 | 176,248.923 | 57,356.914 | 1.19 | 2.88 | 0.29 | 388 | 34 | CS |
| L2P9 | 176,221.675 | 57,363.543 | 0.88 | 2.84 | 0.62 | 388 | 34 | CW |
| L2P10 | 176,194.428 | 57,370.171 | 0.55 | 3.44 | 0.76 | 388 | 28 | NCS |
| L2P11 | 176,167.181 | 57,376.8 | 0.17 | 3.44 | 8.9 | 388 | 28 | NCS |
| L2P12 | 176,139.934 | 57,383.428 | −0.20 | 2.91 | 0.68 | 388 | 33 | CW |
| L2P13 | 176,112.686 | 57,390.056 | −0.55 | 3.69 | 0.71 | 388 | 26 | NCS |
| L2P14 | 176,085.439 | 57,396.685 | −0.89 | 3 | 0.2 | 388 | 32 | CS |
| L2P15 | 176,058.192 | 57,403.313 | −1.27 | 3 | 0.48 | 388 | 32 | NCS |
| L4P4 | 176,244.645 | 56,849.457 | 9.94 | 7.75 | 0.81 | 374 | 12 | CW |
| L4P5 | 176,216.452 | 56,856.119 | 3.96 | 6.47 | 3.79 | 388 | 15 | CW |
| L4P6 | 176,188.259 | 56,862.781 | 2.41 | 5.84 | 0.25 | 388 | 17 | CS |
| L4P7 | 176,160.066 | 56,869.443 | 1.48 | 15.56 | 0.21 | 388 | 6 | NCS |
| L4P8 | 176,131.873 | 56,876.106 | 1.07 | 7.44 | 0.99 | 388 | 13 | CW |
| L4P11 | 176,047.295 | 56,896.092 | 0.07 | 6.22 | 0.46 | 388 | 16 | NCS |
| L4P12 | 176,019.103 | 56,902.753 | −0.27 | 5.94 | 0.2 | 388 | 16 | CS |
| L4P13 | 175,990.91 | 56,909.416 | −0.75 | 5.59 | 0.15 | 388 | 17 | CS |
| L4P14 | 175,962.718 | 56,916.078 | −1.27 | 4.81 | 0.74 | 388 | 20 | CS |
| L4P15 | 175,934.525 | 56,922.739 | −1.76 | 4.84 | 1 | 388 | 20 | CW |
| L6P1 | 176,202.844 | 56,345.566 | 31.09 | 8.13 | 3.14 | 374 | 12 | NCS |
| L6P2 | 176,174.751 | 56,352.123 | 24.76 | 6.03 | 2.3 | 374 | 16 | NCS |
| L6P3 | 176,146.659 | 56,358.681 | 11.62 | 17.88 | 4.74 | 374 | 5 | CW |
| L6P4 | 176,118.566 | 56,365.238 | 4.30 | 8.13 | 0.2 | 388 | 12 | CS |
| L6P5 | 176,090.473 | 56,371.795 | 2.42 | 7.03 | 1.99 | 388 | 14 | CW |
| L6P6 | 176,062.381 | 56,378.353 | 1.53 | 6.19 | 0.55 | 388 | 16 | CW |
| L6P7 | 176,034.289 | 56,384.91 | 1.05 | 6.34 | 2.77 | 388 | 15 | CW |
| L6P8 | 176,006.196 | 56,391.468 | 0.74 | 6.03 | 1.33 | 388 | 16 | CW |
| L6P9 | 175,978.104 | 56,398.026 | 0.27 | 5.81 | 1.39 | 388 | 17 | CW |
| L6P10 | 175,950.012 | 56,404.584 | −0.05 | 5.47 | 1.39 | 388 | 18 | CW |
| L6P11 | 175,921.919 | 56,411.141 | −0.23 | 5.31 | 0.02 | 388 | 18 | CS |
| L6P12 | 175,893.827 | 56,417.699 | −0.50 | 4.97 | 0.39 | 388 | 20 | CW |
| L6P13 | 175,865.735 | 56,424.257 | −1.18 | 4.84 | 0.04 | 388 | 20 | CS |
| L6P14 | 175,837.642 | 56,430.814 | −1.74 | 4.69 | 0.25 | 388 | 21 | CS |
| L8P2 | 175,980.469 | 55,886.597 | 2.79 | 7.06 | 1.1 | 388 | 14 | CS |
| L8P3 | 175,953.072 | 55,893.172 | 1.78 | 9.09 | 3.3 | 388 | 11 | CS |
| L8P4 | 175,925.676 | 55,899.746 | 1.22 | 7.91 | 4.4 | 388 | 12 | CW |
| L8P6 | 175,870.882 | 55,912.896 | 0.52 | 6.88 | 1.61 | 388 | 14 | CW |
| L8P8 | 175,816.089 | 55,926.046 | −0.17 | 7.38 | 6.13 | 388 | 13 | CW |
| L8P9 | 175,788.693 | 55,932.62 | −0.57 | 6.75 | 0.28 | 388 | 14 | CW |
| L8P10 | 175,761.296 | 55,939.195 | −1.16 | 6.03 | 0.31 | 388 | 16 | NCS |

**Table A1.** *Cont.*

| Code | Easting (OSGB_36) | Northing (OSGB_36) | Elev. (m) | f0 (Hz) | std (Hz) | Vs (m/s) | Beach Depth (m) | Type |
|---|---|---|---|---|---|---|---|---|
| L9P2 | 175,915.633 | 55,639.48 | 2.80 | 7.94 | 2.72 | 388 | 12 | CS |
| L9P3 | 175,889.698 | 55,648.064 | 2.03 | 8.28 | 1.86 | 388 | 12 | CW |
| L9P4 | 175,863.762 | 55,656.648 | 1.45 | 7.75 | 3.54 | 388 | 13 | CW |
| L9P5 | 175,837.827 | 55,665.231 | 1.07 | 8.41 | 0.21 | 388 | 12 | CW |
| L9P8 | 175,760.02 | 55,690.984 | −0.01 | 6.25 | 0.28 | 388 | 16 | CW |
| L9P10 | 175,708.149 | 55,708.15 | −0.69 | 7.22 | 19.85 | 388 | 13 | CW |
| L9P11 | 175,682.214 | 55,716.735 | −1.25 | 6.41 | 0.95 | 388 | 15 | CW |
| L9P12 | 175,656.278 | 55,725.319 | −1.64 | 5.63 | 0.75 | 388 | 17 | CW |
| L10P2 | 175,777.273 | 55,424.628 | 1.15 | 8.38 | 1.6 | 388 | 12 | CS |
| L10P3 | 175,751.601 | 55,433.334 | 0.71 | 11 | 6.24 | 388 | 9 | CW |
| L10P4 | 175,725.93 | 55,442.04 | 0.38 | 9.59 | 0.75 | 388 | 10 | CW |
| L10P6 | 175,674.586 | 55,459.452 | −0.11 | 6.78 | 0.56 | 388 | 14 | CW |
| L10P7 | 175,648.915 | 55,468.157 | −0.41 | 6.13 | 4.13 | 388 | 16 | CW |
| L10P8 | 175,623.243 | 55,476.862 | −0.85 | 5.75 | 0.12 | 388 | 17 | CS |
| L11P3 | 175,706.288 | 55,191.129 | 1.08 | 8.59 | 0.29 | 388 | 11 | CS |
| L11P4 | 175,679.054 | 55,200.28 | 0.42 | 10.19 | 2.62 | 388 | 10 | CW |
| L11P6 | 175,624.585 | 55,218.585 | 0.00 | 8.06 | 2.54 | 388 | 12 | CW |
| L11P7 | 175,597.351 | 55,227.736 | −0.22 | 6.63 | 1.1 | 388 | 15 | CS |
| L11P8 | 175,570.117 | 55,236.888 | −0.61 | 5.78 | 0.33 | 388 | 17 | CS |
| L11P9 | 175,542.882 | 55,246.041 | −1.06 | 5.13 | 0.9 | 388 | 19 | CW |
| L11P10 | 175,515.648 | 55,255.193 | −1.47 | 4.38 | 0.35 | 388 | 22 | CS |
| L12P2 | 175,888.448 | 54,867.244 | 24.99 | 5.78 | 0.74 | 374 | 16 | CW |
| L12P3 | 175,861.355 | 54,876.687 | 18.59 | 5.91 | 1.76 | 374 | 16 | CW |
| L12P4 | 175,834.262 | 54,886.131 | 13.56 | 7.22 | 1.79 | 374 | 13 | CS |
| L12P5 | 175,807.169 | 54,895.574 | 7.28 | 6.22 | 1.7 | 374 | 15 | CW |
| L12P6 | 175,780.076 | 54,905.019 | 3.86 | 5.91 | 2.76 | 388 | 16 | CW |
| L12P7 | 175,752.983 | 54,914.462 | 3.11 | 5.44 | 0.55 | 388 | 18 | CW |
| L12P8 | 175,725.89 | 54,923.905 | 2.40 | 4.34 | 1.96 | 388 | 22 | CS |
| L12P9 | 175,698.797 | 54,933.349 | 1.93 | 4.06 | 0.36 | 388 | 24 | CW |
| L12P10 | 175,671.704 | 54,942.792 | 1.55 | 4.16 | 0.91 | 388 | 23 | CS |
| L12P12 | 175,617.517 | 54,961.679 | 0.95 | 4 | 0.83 | 388 | 24 | CW |
| L12P14 | 175,563.331 | 54,980.566 | 0.33 | 5.91 | 0.09 | 388 | 16 | CW |
| L12P17 | 175,482.052 | 55,008.897 | −0.71 | 5.5 | 0.8 | 388 | 18 | NCS |
| L12P18 | 175,454.959 | 55,018.341 | −1.07 | 5.28 | 0.15 | 388 | 18 | CS |
| L12P19 | 175,427.866 | 55,027.784 | −1.50 | 5.16 | 1.87 | 388 | 19 | CW |
| L13P1 | 176,034.296 | 54,611.834 | 30.66 | 7.38 | 5.16 | 374 | 13 | CW |
| L13P4 | 175,950.637 | 54,640.19 | 16.89 | 7.28 | 1.24 | 374 | 13 | CW |
| L13P5 | 175,922.751 | 54,649.643 | 12.97 | 5.59 | 3.46 | 374 | 17 | NCS |
| L13P6 | 175,894.864 | 54,659.094 | 11.16 | 2.84 | 0.72 | 374 | 33 | NCS |
| L13P7 | 175,866.978 | 54,668.547 | 12.01 | 3.13 | 0.41 | 374 | 30 | CS |
| L13P8 | 175,839.092 | 54,677.999 | 13.53 | 3.56 | 0.61 | 374 | 26 | CS |
| L13P9 | 175,811.205 | 54,687.451 | 13.69 | 3.75 | 1.05 | 374 | 25 | CS |
| L13P10 | 175,783.319 | 54,696.904 | 5.99 | 3.81 | 0.38 | 374 | 25 | CS |
| L13P11 | 175,755.432 | 54,706.356 | 6.37 | 4.78 | 1.83 | 374 | 20 | CS |
| L13P12 | 175,727.546 | 54,715.808 | 4.49 | 5 | 2.29 | 388 | 19 | NCS |
| L13P13 | 175,699.66 | 54,725.26 | 3.46 | 4.81 | 0.14 | 388 | 20 | CW |
| L13P14 | 175,671.773 | 54,734.712 | 2.68 | 5.63 | 1.01 | 388 | 17 | CW |
| L13P15 | 175,643.887 | 54,744.164 | 2.10 | 6.53 | 0.53 | 388 | 15 | NCS |
| L13P16 | 175,616.001 | 54,753.616 | 1.67 | 3.25 | 0.39 | 388 | 30 | CW |
| L13P18 | 175,560.228 | 54,772.521 | 1.13 | 8.03 | 4.77 | 388 | 12 | NCS |
| L13P19 | 175,532.342 | 54,781.973 | 0.84 | 7.31 | 0.06 | 388 | 13 | CW |
| L13P20 | 175,504.455 | 54,791.425 | 0.49 | 6.47 | 3.69 | 388 | 15 | CS |
| L13P21 | 175,476.569 | 54,800.878 | 0.18 | 5.34 | 0.04 | 388 | 18 | CS |
| L13P22 | 175,448.682 | 54,810.33 | −0.18 | 5.19 | 1.95 | 388 | 19 | CS |
| L13P23 | 175,420.796 | 54,819.782 | −0.53 | 5.19 | 2.42 | 388 | 19 | CW |
| L13P24 | 175,392.91 | 54,829.234 | −0.94 | 4.97 | 2.12 | 388 | 20 | CS |
| L13P25 | 175,365.023 | 54,838.686 | −1.36 | 4.84 | 0.42 | 388 | 20 | CW |

**Table A1.** *Cont.*

| Code | Easting (OSGB_36) | Northing (OSGB_36) | Elev. (m) | f0 (Hz) | std (Hz) | Vs (m/s) | Beach Depth (m) | Type |
|---|---|---|---|---|---|---|---|---|
| L14P2 | 175,721.267 | 54,503.42 | 3.70 | 24.5 | 8.26 | 388 | 4 | CS |
| L14P3 | 175,693.359 | 54,514.423 | 3.63 | 12.97 | 3.72 | 388 | 7 | NCS |
| L14P4 | 175,665.451 | 54,525.426 | 3.28 | 5.19 | 0.28 | 388 | 19 | CS |
| L14P5 | 175,637.543 | 54,536.428 | 2.84 | 5.09 | 0.14 | 388 | 19 | CS |
| L14P6 | 175,609.635 | 54,547.431 | 2.44 | 4.94 | 1.48 | 388 | 20 | CS |
| L14P8 | 175,553.818 | 54,569.436 | 1.93 | 6.78 | 15.5 | 388 | 14 | CW |
| L14P14 | 175,386.37 | 54,635.452 | −0.05 | 6.91 | 2.24 | 388 | 14 | CS |
| L14P15 | 175,358.462 | 54,646.456 | −0.38 | 6.56 | 1.84 | 388 | 15 | CS |
| L14P16 | 175,330.554 | 54,657.458 | −0.77 | 6.31 | 4 | 388 | 15 | CS |
| L14P17 | 175,302.646 | 54,668.461 | −1.22 | 6.09 | 0.16 | 388 | 16 | CS |
| L14P18 | 175,274.738 | 54,679.464 | −1.65 | 6.13 | 0.1 | 388 | 16 | NCS |
| L15P2 | 175,319.169 | 54,472.855 | −0.18 | 10 | 0.64 | 388 | 10 | CW |
| L15P3 | 175,293.452 | 54,488.125 | −0.56 | 8.75 | 6.37 | 388 | 11 | CS |
| L15P4 | 175,267.736 | 54,503.393 | −1.05 | 8.03 | 3.76 | 388 | 12 | CS |
| L15P5 | 175,242.02 | 54,518.661 | −1.46 | 7.72 | 1.94 | 388 | 13 | CS |
| L15P6 | 175,216.303 | 54,533.93 | −1.89 | 7.38 | 0.09 | 388 | 13 | CS |

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
