# Peer review of "Evidence of Former Sea Levels from a Passive Seismic Survey at a Sandy Beach; Perranporth, SW England, UK"

_jmse, doi:10.3390/jmse10050569_

Round 1
Reviewer 1 Report
Comments to the paper by Payo and co-workers entitled “Evidence of former sea levels from a Passive Seismic Survey at a sandy beach; Perranporth, SW England, UK”
This paper deals with a routine application of “seismic noise” method to determine the soil structure in order to find a method alternative to the ordinary geological observations to infer the former sea levels at a SW England site.
The paper is well structured, clearly written and interesting. Method, yet ordinary, is sufficiently reported. Results are interesting, and the final considerations, not conclusive anyway, favor future attempts to complement the ordinary studies to reconstruct the sea level with seismic observations.
In synthesis, the paper deserves publication as such.
Reviewer 2 Report
Review of “Evidence of former sea levels from a Passive Seismic Survey at a sandy beach; Perranporth, SW England, UK” by Andres Payo et al.
This paper presents a comprehensive study on estimating the thickness of sediments at beach, further providing evidence of former sea levels. The data collection, processing and interpretation are pretty good and well written. I have only three moderate points to the authors:
- Figure 2: Please place grid with longitude and latitude labels to the map.
- Theoretically, the predominant frequency (f0) is a function of both Vs and thickness of sediments (h). However, many studies summarize empirical regression model between f0 and h, based on which we can rapidly estimate h for any given f0. Therefore, I would suggest the authors to estimate h using the models as summarized in e.g. Guo et al., 2021.
Guo, Z., A. Aydin, Y. Huang, M. Xue, 2021. Polarization characteristics of Rayleigh waves to improve seismic site effects analysis by HVSR method, Engineering Geology, 292, 106274.
- Figure 9 and 10: The back theory of HVSR modeling should be briefly introduced.
